# Suspected Cerebral Salt Wasting Syndrome with Cervical Spinal Lesion in a Domestic Shorthair Cat

**DOI:** 10.3390/vetsci10060385

**Published:** 2023-06-01

**Authors:** Minkun Kim, Woo-Jin Song, Jongjin Park, Saeyoung Lee, Sangkyung Choen, Myung-Chul Kim, Youngmin Yun

**Affiliations:** 1Laboratory of Veterinary Internal Medicine, College of Veterinary Medicine, Jeju National University, Jeju 63243, Republic of Korea; ketopi92@naver.com (M.K.); pjj4024@gmail.com (J.P.); qr3397@naver.com (S.L.); dvmyun@jejunu.ac.kr (Y.Y.); 2The Research Institute of Veterinary Science, College of Veterinary Medicine, Jeju National University, Jeju 63243, Republic of Korea; mck@jejunu.ac.kr; 3Department of Surgical and Radiological Sciences, School of Veterinary Medicine, University of California, Davis, CA 95616, USA; chunsk1987@gmail.com; 4Diagnostic Laboratory Medicine, College of Veterinary Medicine, Jeju National University, Jeju 63243, Republic of Korea

**Keywords:** cerebral salt wasting syndrome, cat, fludrocortisone, hyponatremia, ischemic myelopathy

## Abstract

**Simple Summary:**

Cerebral salt wasting syndrome is defined as a hyponatremic condition due to sodium loss through the kidney during intracranial disorders. A 12-year-old spayed female domestic shorthair cat was referred to our referral animal hospital with tetraplegia and a history of hyponatremia with dehydration from a local hospital after intravenous fluid treatment. The cat was suspected to have intracranial disease after a thorough physical and neurologic examination. MRI revealed bilateral parietal lobe cerebrocortical gray-white matter junction T2/Flair high signal related to rapid electrolyte calibration, and C2 spinal cord ventral area T2 high signal indicated ischemic myelopathy. After 2 days, the cat revisited our hospital with severe lethargy and anorexia. Laboratory examination revealed hyponatremia (136 mEq/L) with dehydration. Through history-taking, laboratory examination, imaging, and fluid therapeutic response, the cerebral salt wasting syndrome was tentatively diagnosed. Electrolytes stabilized after starting treatment with fludrocortisone, and the cat was discharged. After 3 weeks, MRI revealed that the parietal lobe lesion had disappeared; however, the spinal cord ischemic lesion got worse than previous imaging, and the cat was euthanized. This is the first case of suspected cerebral salt-wasting syndrome in a cat. And this syndrome might be included on the differential diagnosis list for cats with intracranial disease, dehydration, and hyponatremia.

**Abstract:**

A 12-year-old spayed female domestic short cat was presented with tetraplegia. The cat also showed signs of hyponatremia and dehydration, which were rapidly corrected by intravenous fluid infusion. Based on thorough physical and neurological examinations, the patient was suspected of having an intracranial disease. MRI revealed a high-signal T2 image of the bilateral parietal cerebral cortical gray matter junction, which is associated with fast electrolyte calibration, and a high-signal T2 image of the C2 spinal cord ventral area, which is associated with ischemic myelopathy. The cat reappeared three days later due to anorexia. Laboratory examinations revealed that the cat was clinically dehydrated and exhibited hyponatremia. Other causes of hyponatremia were excluded through history-taking, laboratory examination, imaging, and therapeutic response to fluid therapy, except for cerebral salt-wasting syndrome (CSWS). The cat was discharged 3 days after the start of fludrocortisone therapy with electrolytes within the normal range. Magnetic resonance imaging (MRI) was performed again 1 month after hospitalization, and the cerebral lesion disappeared, but the spinal cord lesion worsened compared to the previous image. The patient was euthanized due to the progression of the spinal lesion, with a poor prognosis and poor quality of life. This is the first case of suspected CSWS with a cervical spinal lesion in a cat.

## 1. Introduction

Hyponatremia is defined as a sodium concentration below the reference interval and is caused by an excess of water relative to sodium in the extracellular fluid [1,2]. Clinical signs can range from asymptomatic to severe neurological symptoms, including seizures and coma [2]. In hyponatremic animals, it is important to evaluate underlying disorders, and a diagnostic approach for small animals with hyponatremia is based on an assessment of both osmolality and volemic state [1].

In human medicine, cerebral salt wasting syndrome (CSWS) is defined as sodium loss through the kidney during intracranial disorders, leading to hyponatremia and a decrease in extracellular fluid volume [3]. The pathogenesis of CSWS is not fully understood, but it is known as primary natriuresis, which leads to hypovolemia and sodium depletion without the stimulation of sodium excretion. Natriuretic peptides regulate the water and sodium content of the brain and cerebrospinal fluid production. Brain natriuretic peptide (BNP) may be related to CSWS [4]. In addition, severe extracellular volume expansion could also downregulate transporters involved in renal sodium absorption, and a decrease in renal sympathetic activity could cause an increase in renal flow and glomerular filtration, a decrease in renin release, and a decrease in renal tubular sodium reabsorption [5]. Although the pathogenesis and occurrence of CSWS are controversial, its management is well established [3,6]. Here we describe a case of suspected CSWS in a cat with hyponatremia and a cervical spine lesion.

## 2. Case Description

A 12-year-old spayed female domestic short-haired cat presented with sudden onset of tetraplegia to our referral animal hospital (Day 0). The cat had received fluid therapy for hyponatremia (137 mEq/L; reference interval [RI], 145–165 mEq/L) at a local animal hospital the day before (Day—1). Physical examination revealed a mild systolic murmur (2/6 grade) without a gallop sound, and Echocardiography revealed a hypertrophic cardiomyopathy (HCM) phenotype at the American College of Veterinary Internal Medicine (ACVIM) stage B1, characterized by a thickened diastolic left ventricle free wall (6.6 mm) with a normal diastolic left atrium diameter (9.2 mm). Further tests were performed, including serum concentrations of glucose, lactate, and potassium from each limb, and results were within the normal ranges, indicating a low probability of arterial thromboembolism. Complete blood count (CBC; ProCyte Dx Hematology Analyzer; IDEXX Laboratories, Westbrook, ME, USA) and serum chemistry (Catalyst One Chemistry Analyzer; IDEXX Laboratories), including the concentration of total T4, showed no remarkable findings. However, the moderate elevation of serum amyloid A (SAA) (41.5 μg/dL; RI, 0–5 μg/dL; V-check V200, Bionote, Hwaseong, Republic of Korea) was detected. Interestingly, the serum sodium concentration (163 mEq/L) was within the normal range; indicating that rapid correction (increase > 8 mEq/L at 24 h) was not required (Table 1). No remarkable findings were identified on radiography or abdominal ultrasonography.

Therefore, magnetic resonance imaging (MRI) was performed for the cat with tetraplegia. MRI revealed T2 hyperintensity in regions of the bilateral parietal lobe area (Figure 1A) and T2 hyperintensity in regions of the C2 ventral area (Figure 1B). In addition, there was no significant abnormality in diffusion-weighted imaging of the brain or cervical region. Myelinolysis of the brain lesion due to rapid electrolyte calibration and ischemic myelopathy (IM) of the cervical lesion were tentatively diagnosed. IM due to the HCM phenotype was suspected to be the main cause of tetraplegia, which could be expected to be self-limiting [7]. The cat was scheduled to undergo a rechecking every week for monitoring without medication.

The cat revisited our hospital with worsening anorexia and depression on day 4. Moderate dehydration (7–8%) was suspected on physical examination. In addition, hyponatremia (136 mEq/L) and hypokalemia (3 mEq/L; RI, 3.8–5.0 mEq/L) were detected. No other remarkable findings through CBC and serum chemistry were shown; therefore, an electrolyte imbalance was suspected as the cause of anorexia and depression.

Hypoosmolar hyponatremia with normovolemia or hypovolemia was suspected in hyponatremic animals (Table 2). In addition, other states were ruled out by routine screening tests, including blood analysis, urinalysis (Table 1), and imaging, except for the syndrome of inappropriate antidiuretic hormone (SIADH) and cerebral salt wasting syndrome (CSWS). A therapeutic trial of fluid infusion with 0.45% saline and 2.5% dextrose was prescribed, and the sodium concentration was gradually increased (Figure 2). CSWS was tentatively diagnosed because SIDAH can only be managed by fluid restriction and not by fluid infusion.

On Day 6, fluid infusion was discontinued and fludrocortisone (Florinef tab; Samil Pharm, Seoul, Republic of Korea; 0.02 mg/kg/day per oral) was prescribed according to the human CSWS management protocol [8]. Clinical signs, including depression and anorexia, improved. Serum sodium concentration was still within the reference range on Day 8, and the cat was discharged. After 10 days (Day 18), the serum sodium concentration was normal, and the same dosage of fludrocortisone was prescribed (Figure 2). However, the cat’s tetraplegia did not improve until Day 32, and an MRI was re-performed to monitor the previous lesions in the brain (Figure 1C) and cervical spinal cord (Figure 1D). Lesions in the bilateral parietal lobes had disappeared due to the correction of the electrolyte imbalance; however, the lesions in the cervical spinal region had progressed. Therefore, chronic IM with a poor prognosis was tentatively diagnosed [7,9], and the cat was euthanized on Day 33.

## 3. Discussion

The first diagnostic approach for patients with hyponatremia is to measure the osmotic pressure of the extracellular fluid (ECF). Therefore, hyponatremic states can be divided into 3 types as follows: normal osmolality, hyperosmolality, and hypoosmolality [1]. Normal osmolar hyponatremia and pseudohyponatremia are synonymous, and low serum sodium concentration is found in patients with normal volume status and normal osmotic pressure, which can be induced by hyperproteinemia or hyperlipidemia [10]. The causes of normosmolar hyponatremia could be excluded if plasma protein (7.2 g/dL; RI, 5.7–8.9 g/dL) and plasma total cholesterol (123 mg/dL; RI, 65–225 mg/dL) levels were within the reference interval. Additionally, hyperosmolar hyponatremia is caused by osmotic gradients in the ECF that lead to water movement, including hyperglycemia and mannitol administration [11]. The patients were excluded based on their medical history and serum glucose levels.

Hypoosmolar hyponatremia is divided into three categories depending on the volemic status: hypervolemia, normovolemia, and hypovolemia. The causes of hypervolemic states, including loss of liver function, kidney function, and congestive heart failure [12], were excluded through laboratory examination and imaging. Causes of normovolemic states include primary polydipsia, diuretic application, hypothyroidism, and syndrome of inappropriate secretion of antidiuretic hormone (SIADH), which were excluded by history-taking, medical history, and normal serum total T4 levels (0.8 μL/dL; RI, 0.8–4.7 μL/dL) except SIADH. The causes of hypovolemic states include burn history, hypoadrenocorticism, pancreatitis, peritonitis, gastrointestinal fluid loss, and CSWS, which is the state of salt wasting through the urine due to temporary central nervous system problems with polyuria [1]. The causes of hypoveolemic hypotaremia in the cat were excluded by history-taking, imaging, and a normal basal serum cortisol level (5.91 μL/dL, RI, 0.5–10 μL/dL) except CSWS.

The differences between CSWS and SIADH are shown in Table 3. In brief, the criteria for the diagnosis of SIADH in veterinary medicine include [13]: (1) hyponatremia with hypoosmolality; (2) high urine osmolality; (3) normal renal, adrenal, and thyroid function; (4) presence of natriuresis; (5) no evidence of hypovolemia, ascites, or edema; and (6) correction by fluid restriction. However, the diagnostic criteria for CSWS have not yet been established in veterinary medicine. In human medicine, CSWS can be included in the differential diagnosis of hyponatremia in patients with cerebral insults [14]. Uygun et al. were the first to propose the following CSWS diagnostic criteria [15]: (1) central nervous system injury, (2) plasma [Na] 130 mmol/L, (3) urine [Na] > 80 mmol/day or >20 mmol/L, (4) osmotic pressure of plasma 270 mmol/L, (5) urine osmotic pressure/blood osmotic pressure >1, and (6) urine volume >1800 mL/day. In addition, Leonard et al. proposed the main criteria to diagnose CSWS [16]: (1) brain pathology examination; (2) hyponatremia; (3) hypovolemia; and (4) urinary salt loss. In this case, the cat had a central nervous system injury (a cervical lesion), marked hyponatremia, and hypovolemia (dehydration). Unfortunately, the urinary salt loss was not evaluated in this study. However, the cat’s hyponatremia improved with fluid infusion and not with fluid restriction, which suggests a tentative diagnosis of CSWS [17]. CSWS can be managed with fluid therapy and fludrocortisone, whereas SIADH can be managed with fluid restriction and the vasopressin V2-receptor antagonist tolvaptan [18]. The cat showed improvement with fluid therapy, and the electrolyte balance was managed with fludrocortisone only, thus suggesting a diagnosis of CSWS.

Fludrocortisone acetate is a mineralocorticoid that prevents plasma volume depletion and maintains sodium and fluid balance [19]. It affects the distal convoluted tubules and collecting ducts, enhancing sodium and water retention and increasing potassium and hydrogen excretion, leading to ECF volume expansion [20]. The use of fludrocortisone to manage CSWS in humans has been previously reported [6]. Fludrocortisone acetate’s starting dosage in dogs and cats is 0.01–0.02 mg/kg/day per oral dose in hypoadrenocorticism and should be adjusted mainly according to the serum potassium concentration [21]. In this case, the cat was tentatively diagnosed with CSWS, and fludrocortisone acetate successfully maintained sodium and fluid balance.

However, the cat was euthanized due to prolonged IM, which induced tetraplegia. Theobald et al. retrospectively reported 19 cases, and approximately 42% of cats were euthanized because of a poor response [22]. Chronic kidney disease, hypertension, cardiomyopathy, and hyperthyroidism have all been suspected as causes of ischemic damage to the spinal cord [23]. The cat in this case was diagnosed with the HCM phenotype, ACVIM stage B1. Also, the cat was expected to have a poor prognosis through an MRI recheck despite the presence of well-controlled electrolytes due to CSWS.

However, this case report has some limitations. First, we could not evaluate all the diagnostic values of CSWS criteria in human medicine, including urine fractional excretion of sodium, urine osmotic pressure, serum renin, and aldosterone concentrations [24,25]. However, we ruled out other differential diagnoses of hyponatremia, including SIADH, in the cats. Second, we could not demonstrate that cervical spinal cord lesions were the exact cause of CSWS in this case. Third, the cat could not undergo cerebrospinal fluid (CSF) analysis for infectious agents, including feline infectious peritonitis virus and *Toxoplasma gondii*.

## 4. Conclusions

This is the first reported case of suspected CSWS in a cat in which hyponatremia was successfully managed using fludrocortisone with IM. In small animal practice, CSWS should be considered a differential diagnosis in cases of hypoosmolar hyponatremia with hypovolemia.

## Figures and Tables

**Figure 1 vetsci-10-00385-f001:**
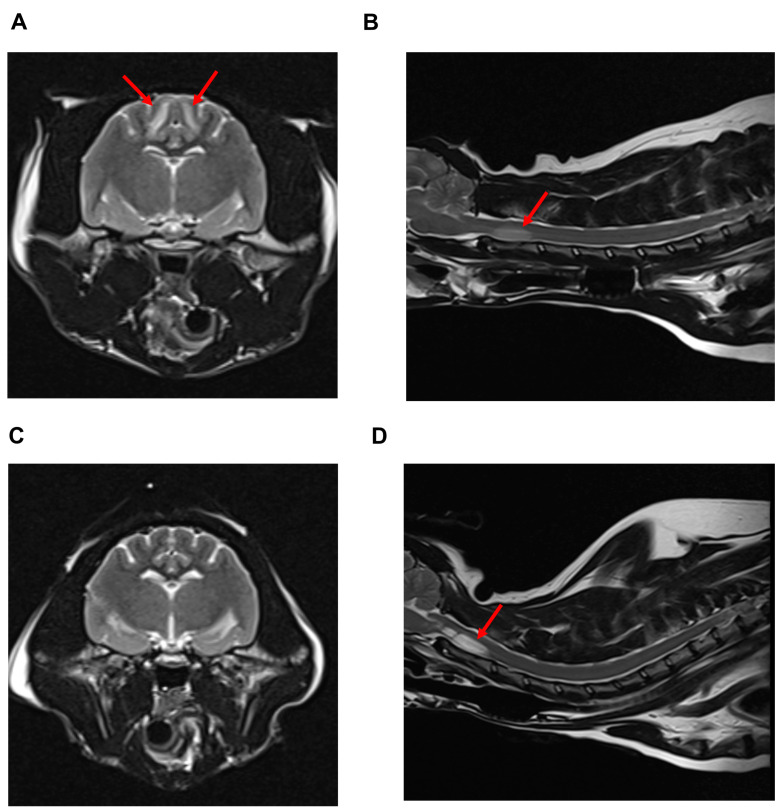
MRI images in this case at Day 1 (**A**,**B**) and Day 32 (**C**,**D**). On day 1, T2 hyperintensity regions of the bilateral parietal lobe area (**A**), red arrows)) and regions of the C2 ventral area (**B**), red arrow) were detected. At day 32, lesions in the bilateral parietal lobe area disappeared after electrolyte correction (**C**); however, lesions in the C2 ventral area progressed (**D**), red arrow)).

**Figure 2 vetsci-10-00385-f002:**
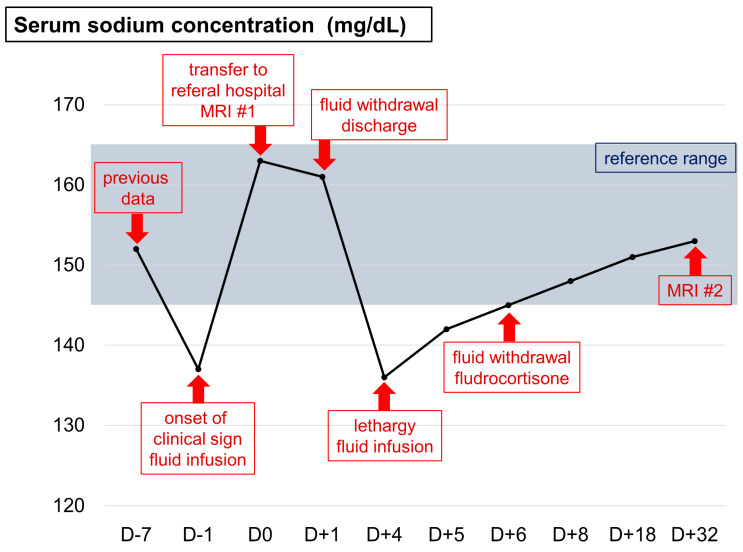
Flow chart of serum sodium concentration in this case. (MRI #1 = the first MRI examination; MRI #2 = the second MRI examination; D = day).

**Table 1 vetsci-10-00385-t001:** Blood analysis results on the day of symptom onset (Day-1).

Parameters	Value	Reference Range	Unit
CBC			
RBC	8.5	6.54–12.2	10^12^/L
HCT	34.9	30.3–52.3	%
#Reticulocytes	18.7	3–50	10^3^/µl
WBC	10.36	2.87–17.02	10^9^/L
PLT	169	151–600	10^9^/L
Serum chemistry			
Na	137	150–165	mmol/L
K	2.9	3.5–5.8	mmol/L
Total protein	7.2	5.7–8.9	g/dL
BUN	27	16–36	mg/dL
Glucose	170	7–159	mg/dL
SDMA	10	0–14	µg/dL
Total T4	0.8	0.8–4.7	µg/dL
Basal cortisol	5.91	0.5–10	µg/dL
SAA	41.5	0–5	µg/dL
NT-proBNP	478	0–100	pmol/L
Coagulation			
D-dimer	<0.1	0–0.1	µg/ml

RBC, red blood cell; HCT, hematocrit; #Reticulocytes = absolute number of reticulocytes; WBC, white blood cell; PLT, platelet; BUN, blood urea nitrogen; SDMA, symmetric dimethylarginine; SAA, serum amyloid A; NT-proBNP, N-terminal prohormone of brain natriuretic peptide.

**Table 2 vetsci-10-00385-t002:** Differential diagnosis list for hyponatremia in cats [1,2].

Hyperosmolality	Normosmolality	Hyposmolality		
		Hypervolemia	Normovolemia	Hypovolemia
Hyperglycemia	Hyperlipidemia	Liver function loss	Primary polydipsia	Hypo- adrenocorticism
Mannitol infusion	Hyperproteinemia	Kidney function loss	Hypothyroidism	Pancreatitis
		Congestive heart failure	Iatrogenic (Diuretics)	Peritonitis
			Syndrome of inappropriate antidiuretic hormone	Gastrointestinal fluid loss
				Burn
				Cerebral salt wasting syndrome

**Table 3 vetsci-10-00385-t003:** Differences between SIADH and CSWS [13,14,15].

Parameters	SIADH	CSWS
Hydration status	Hydrated	Dehydrated
Urine volume	Variable	Increased
Plasma BNP	Variable	Increased
ADH	Increased	Decreased
Serum uric acid	Decreased	Increased
CVP	Increased	Decreased
Management	Fluid restriction	Fluid infusion

SIADH, inappropriate antidiuretic hormone syndrome; CSWS, cerebral salt wasting syndrome; BNP, brain natriuretic protein; ADH, antidiuretic hormone; CVP, central venous pressure.

## Data Availability

The data presented in this study are contained within the article.

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
