# Peer review of "Suspected Cerebral Salt Wasting Syndrome with Cervical Spinal Lesion in a Domestic Shorthair Cat"

_vetsci, 2023, doi:10.3390/vetsci10060385_

Round 1
Reviewer 1 Report
Comments on the manuscript “Suspected Cerebral Salt Wasting Syndrome With Cervical Spinal Lesion in a Domestic Short Hair Cat” submitted to the Veterinary Sciences
General comments
I thank the editor for the opportunity to comment on the manuscript entitled “Suspected Cerebral Salt Wasting Syndrome With Cervical Spinal Lesion in a Domestic Short Hair Cat”.
A rare well-described case of the CSWS in an elderly shorthair cat is presented. The animal had bilateral parietal cerebral inflammation and a cervical spine lesion. Despite the relative success of the therapeutic procedures, the cat was euthanized due poor prognosis, after 32 days.
Like the authors, I am not aware of another article reporting the syndrome in cats in the scientific literature. Thus, it is an original and relevant article to understand hydro electrolytic syndromes in animals.
There is an elegance in the presentation of the case, as can be seen in Figure 2.
The manuscript has some minor errors, but overall it is well-written.
My comments in detail are as follows:
Title
It is a shorthair cat, not “short hair”.
Simple Summary
Line 17: “short cat”, please standardize in the text to “shorthair” cat. The words “referral” and “tetraphelgia” is wrong.
Abstract
It is good.
Introduction
Line 67: “To the best of our knowledge,…” it is a conclusion. In this part of the text you should write the study objective.
Case Description
Table 1: As was written in Table 3, the authors must write in the sub-caption what each abbreviation of the clinical parameters means.
Figure 2: d= day
Conclusions
It is good.
References
It is good.
Author Response
Reviewer #1
I thank the editor for the opportunity to comment on the manuscript entitled “Suspected Cerebral Salt Wasting Syndrome With Cervical Spinal Lesion in a Domestic Short Hair Cat”.
A rare well-described case of the CSWS in an elderly shorthair cat is presented. The animal had bilateral parietal cerebral inflammation and a cervical spine lesion. Despite the relative success of the therapeutic procedures, the cat was euthanized due poor prognosis, after 32 days.
Like the authors, I am not aware of another article reporting the syndrome in cats in the scientific literature. Thus, it is an original and relevant article to understand hydro electrolytic syndromes in animals.
There is an elegance in the presentation of the case, as can be seen in Figure 2.
The manuscript has some minor errors, but overall it is well-written.
My comments in detail are as follows:
1. COMMNET: Title, It is a shorthair cat, not “short hair”.
RESPONSE: Thank you for your detailed comment. We have revised the title as your comment.
2. COMMENT: Simple Summary, Line 17: “short cat”, please standardize in the text to “shorthair” cat. The words “referral” and “tetraphelgia” is wrong.
RESPONSE: Thank you for your detailed comment, and we have revised as your comment.
3. COMMENT: Abstract, It is good.
RESPONSE: Thank you for your kind comment.
4. COMMENT: Introduction, Line 67: “To the best of our knowledge,…” it is a conclusion. In this part of the text you should write the study objective.
RESPONSE: Thank you for your detailed comment. We have revised the sentence as follow: “Here we describe a case of suspected CSWS in a cat with hyponatremia and a cervical spine lesion”.
5. COMMENT: Case Description, Table 1: As was written in Table 3, the authors must write in the sub-caption what each abbreviation of the clinical parameters means.
RESPONSE: Thank you for your detailed comment. We have added sub-caption in Table 1 as your comment.
6. COMMENT: Figure 2: d= day
RESPONSE: Thank you for your detailed comment. We have added it in figure legend as your comment.
7. COMMENT: Conclusions, It is good. References, It is good.
RESPONSE: Thank you for your kind comment.
Reviewer 2 Report
A very interesting case report. Thank you for providing this information about workup of the severe hyponatremia.
In the simple summary there are a number of grammatical and syntax errors detected. Example: line 16 it should read short hair cat and not short cat.
The case report would be much strengthened if the authors would be able to present aldosterone levels and although difficult to fine a laboratory these days to perform species specific renin analysis.
Line 18: should read the cat WAS suspected to have intracranial disease....
Case Description
Line 80: reference 1 needs to be placed in brackets.
Lines 90-97: MRI findings- was any diffusion sequencing performed for further evaluation for the intracranial and cervical lesions? IF so please provide that information.
Although difficult to obtain as few diagnostic laboratories offer species specific renin levels these days the report would benefit from having determination of renin and aldosterone levels in further support the diagnosis of Salt Wasting Syndrome
The normal basal cortisol concentration mentioned in the Discussion LINE157 should be reported in the RESULTS section and in Table 1. New results should not be reported in the discussion session.
Although a thorough diagnostic workup was performed, missing data that might confirmed other etiologies or more strongly support SWS would be renin, aldosterone levels, fractional excretion of Urine Na and determination of urine osmolarity compared to the cat's serum osmolarity.
Reference section: the references have double numbering. Should be corrected.
Given the fact that this is a rare condition it is acceptable for the number of older articles included in the reference list. However, there are a number of 2020-2023 articles in the human literature.
There are a number of syntax and grammatical errors in the simple summary but few in the rest of the manuscript
Author Response
Reviewer #2
A very interesting case report. Thank you for providing this information about workup of the severe hyponatremia.
1. COMMENT: In the simple summary there are a number of grammatical and syntax errors detected. Example: line 16 it should read short hair cat and not short cat.
RESPONSE: Thank you for your detailed comment. We have revised the paragraph in this revision, and we have attached certification of English editing for this manuscript.
2. COMMENT: The case report would be much strengthened if the authors would be able to present aldosterone levels and although difficult to fine a laboratory these days to perform species specific renin analysis.
RESPONSE: Thank you for your detailed comment. As your comment, we could not evaluate all the diagnostic value of CSWS criteria in human medicine, including urine fractional excretion of sodium, urine osmotic pressure, serum renin and aldosterone concentrations (Cui, H. et al., Inappropriate antidiuretic hormone secretion and cerebral salt-wasting syndromes in neurological patients. Front. Neurosci. 2019, 13, 1170.; Bouchlarhem, A. et al., Cerebral Salt Wasting Syndrome (CSW): An unusual cause of hypovolemia after spontaneous cerebral hem-orrhage successfully treated with fludrocortisone. Radiol. Case Rep. 2022, 17(1), 106-110). We have described this limitation in discussion section. We hope our approach acceptable.
3. COMMENT: Line 18: should read the cat WAS suspected to have intracranial disease....
RESPONSE: Thank you for your detailed comment, and we have revised the sentence.
4. COMMENT: Case Description, Line 80: reference 1 needs to be placed in brackets.
RESPONSE: Thank you for your detailed comment, and we deleted the number.
5. COMMENT: Lines 90-97: MRI findings- was any diffusion sequencing performed for further evaluation for the intracranial and cervical lesions? IF so please provide that information.
RESPONSE: Thank you for your valuable comment. MRI revealed T2 hyperintensity in regions of bilateral parietal lobe area and T2 hyperintensity in regions of C2 ventral area. In addition, there was no significant abnormality in diffusion-weighted imaging of the brain and cervical region. Therefore, myelinolysis of the brain lesion due to rapid electrolyte calibration and ischemic myelopathy (IM) of the cervical lesion were tentatively diagnosed. It has been described in the revision.
6. COMMENT: Although difficult to obtain as few diagnostic laboratories offer species specific renin levels these days the report would benefit from having determination of renin and aldosterone levels in further support the diagnosis of Salt Wasting Syndrome
RESPONSE: Thank you for your detailed comment. As your comment, we could not evaluate all the diagnostic value of CSWS criteria in human medicine, including urine fractional excretion of sodium, urine osmotic pressure, serum renin and aldosterone concentrations. We have described this limitation in discussion section. We hope our approach acceptable.
7. COMMNET: The normal basal cortisol concentration mentioned in the Discussion LINE157 should be reported in the RESULTS section and in Table 1. New results should not be reported in the discussion session.
RESPONSE: Thank you for your valuable comment. We have added result of basal cortisol concentration in Table 1.
8. COMMENT: Although a thorough diagnostic workup was performed, missing data that might confirmed other etiologies or more strongly support SWS would be renin, aldosterone levels, fractional excretion of Urine Na and determination of urine osmolarity compared to the cat's serum osmolarity.
RESPONSE: Thank you for your detailed comment. As your comment, we could not evaluate all the diagnostic value of CSWS criteria in human medicine, including urine fractional excretion of sodium, urine osmotic pressure, serum renin and aldosterone concentrations. We have described this limitation in discussion section. We hope our approach acceptable.
9. COMMENT: Reference section: the references have double numbering. Should be corrected.
Given the fact that this is a rare condition it is acceptable for the number of older articles included in the reference list. However, there are a number of 2020-2023 articles in the human literature.
RESPONSE: Thank you for your detailed comment .We have corrected the references. In addition, we have added 2 recent references of human CSWS as your comment (Cui, H. et al., Inappropriate antidiuretic hormone secretion and cerebral salt-wasting syndromes in neurological patients. Front. Neurosci. 2019, 13, 1170.; Bouchlarhem, A. et al., Cerebral Salt Wasting Syndrome (CSW): An unusual cause of hypovolemia after spontaneous cerebral hem-orrhage successfully treated with fludrocortisone. Radiol. Case Rep. 2022, 17(1), 106-110).